# The ATPase Inhibitory Factor 1 (IF1) Contributes to the Warburg Effect and Is Regulated by Its Phosphorylation in S39 by a Protein Kinase A-like Activity

**DOI:** 10.3390/cancers16051014

**Published:** 2024-02-29

**Authors:** José M. Cuezva, Sonia Domínguez-Zorita

**Affiliations:** 1Departamento de Biología Molecular, Centro de Biología Molecular Severo Ochoa, Consejo Superior de Investigaciones Científicas—Universidad Autónoma de Madrid (CSIC—UAM), 28049 Madrid, Spain; 2Centro de Investigación Biomédica en Red de Enfermedades Raras (CIBERER) ISCIII, 28029 Madrid, Spain; 3Instituto de Investigación Hospital 12 de Octubre, Universidad Autónoma de Madrid, 28049 Madrid, Spain

**Keywords:** mitochondria, ATPase Inhibitory Factor 1, ATP synthase, metabolic reprogramming, cancer, oxidative phosphorylation

## Abstract

**Simple Summary:**

The ATPase Inhibitory Factor 1 (IF1) plays a prominent role as an inhibitor of mitochondrial ATP synthase. Herein we review the role of IF1 in promoting the Warburg effect and the relevance of its phosphorylation by protein kinase A (PKA) to block its activity as inhibitor of ATP synthase. Our results are supported by other independent studies and highlight the involvement of IF1 in metabolic reprogramming to promote an augmented glycolytic phenotype under aerobic conditions by exerting its inhibitory action on a fraction of ATP synthase present in the mitochondrion. The phosphorylation in vivo of IF1 at S39 by a mitochondrial c-AMP-dependent PKA activity prevents its interaction with ATP synthase. Finally, we stress that IF1 not only can play a cancer-promoting role but also can prevent metastatic disease in some carcinomas.

**Abstract:**

The relevant role played by the ATPase Inhibitory Factor 1 (IF1) as a physiological in vivo inhibitor of mitochondrial ATP synthase in cancer and non-cancer cells, and in the mitochondria of different mouse tissues, as assessed in different genetic loss- and gain-of-function models of IF1 has been extensively documented. In this review we summarize our findings and those of others that favor the implication of IF1 in metabolic reprogramming to an enhanced glycolytic phenotype, which is mediated by its binding and inhibition of the ATP synthase. Moreover, we emphasize that IF1 is phosphorylated in vivo in its S39 by the c-AMP-dependent PKA activity of mitochondria to render an inactive inhibitor that is unable to interact with the enzyme, thus triggering the activation of ATP synthase. Overall, we discuss and challenge the results that argue against the role of IF1 as in vivo inhibitor of mitochondrial ATP synthase and stress that IF1 cannot be regarded solely as a pro-oncogenic protein because in some prevalent carcinomas, it prevents metastatic disease.

## 1. Introduction

Recent reports state that the pro-oncogenic protein IF1 does not contribute to the Warburg effect and is not regulated by protein kinase A (PKA), questioning several of the results that we and other laboratories have generated in the field. In this contribution, we have decided to provide a short review of our findings (Figure 1, Figure 2, Figure 3, Figure 4, Figure 5 and Figure 6) and possible explanations of the results that question the role of IF1 in metabolic reprogramming. The overall purpose is to contribute to the clarification of the field, so that the scientific community can continue to promote advances in this field of mitochondrial research.

A role for the participation of IF1 in metabolic reprogramming of the cells was ruled out because it was found that after silencing IF1 in 143B osteosarcoma, HCT116 colon cancer, and HeLa cervix adenocarcinoma cell lines, the rates of ATP synthesis, basal cellular respiration, and mitochondrial membrane potential (ΔΨm) resulted in no relevant changes [1]. These findings contradict studies in a large number of cancer and non-cancer cells using IF1-silenced [2,3], IF1 knock-out [4,5], and overexpressing IF1 [2,3], both in transiently and stably transfected cells (Figure 1, Figure 2, Figure 3 and Figure 4). Importantly, the role of IF1 as a physiologically relevant inhibitor of ATP synthesis by the enzyme in vivo has been amply demonstrated in the mitochondria of transgenic mice with tissue-specific regulated expression of IF1 [6,7], or of its mutant IF1-H49K [8,9,10], or in tissue-specific genetic mouse models of loss of function of IF1 [7,11,12]. Following these experimental approaches there is little doubt that IF1 regulates both the ATP synthetic (Figure 2) and hydrolytic activities (Figure 5) of a fraction of ATP synthase in the mitochondrion under normal aerobic conditions.

We should stress that the inhibitory role of IF1 in ATP synthase in vivo is also supported by other laboratories both in cells in vitro [13,14,15], in isolated islet β-cells [16], and in mouse heart in vivo [17]. The latter study is of special relevance because it has documented that the increase in IF1 in pathological cardiac hypertrophy, in agreement with similar findings by others [18], results in the formation of nonproductive ATP synthase tetramers, as assessed by chemical cross-linking mass spectrometry analysis of hypertrophied hearts [17], which also agrees with recent cryo-EM structures of the tetrameric IF1-inhibited ATP synthase isolated from hearts of other mammalian species [19,20]. Moreover, using ATP IF1 gain- and loss-of-function cell models, they demonstrated that the stalled electron flow due to impaired ATP synthase activity triggered mitochondrial ROS generation leading to the activation of glycolysis [17]. Furthermore, these authors showed that cardiac-specific deletion of *Atp5if1* in mice prevented the metabolic switch and protected against the pathological remodeling during chronic stress, strongly supporting a critical role for IF1 in metabolic rewiring during the pathological remodeling of the heart [17], as well as our previous findings in cancer and non-cancer cells and in other mouse tissues. However, these contributions are ignored in the alluded study [1]. In next sections, we comment on the results and experimental designs in [1] to contribute to the clarification of the field.

## 2. The Role of IF1 in Warburg Effect: Evidence from Metabolic Reprogramming

It is generally agreed that the Warburg effect represents the enhanced production of lactate by cells through the fermentation of glucose in the presence of oxygen. As suggested by Warburg [21], following the principles of the Pasteur effect, it results from a limited activity of mitochondrial oxidative phosphorylation to compensate for the supply of cellular ATP [22]. However, in the alluded paper [1], it is claimed that IF1 does not contribute to the Warburg effect but there is no determination of the rates of aerobic glycolysis after silencing the cellular content of IF1 in any of the three cell lines studied. In Figure 1, we summarize the metabolic reprograming to an enhanced glycolytic phenotype of cancer and non-cancer cells when the mitochondrial content of IF1 is increased by its overexpression, both in transiently and in stably transfected cells. Figure 1 also shows the decrease in glycolytic flux when IF1 is silenced or knocked out. Moreover, Figure 1 also summarizes the rates of aerobic glycolysis in primary neuronal cultures derived from the brains of IF1-overexpressing and IF1-knock-out mice, confirming the in vivo role of IF1 in metabolic reprograming to an enhanced glycolytic phenotype. In other words, IF1 contributes to the Warburg effect as assessed by the determination of the rates of aerobic glycolysis (Figure 1).

**Figure 1 cancers-16-01014-f001:**
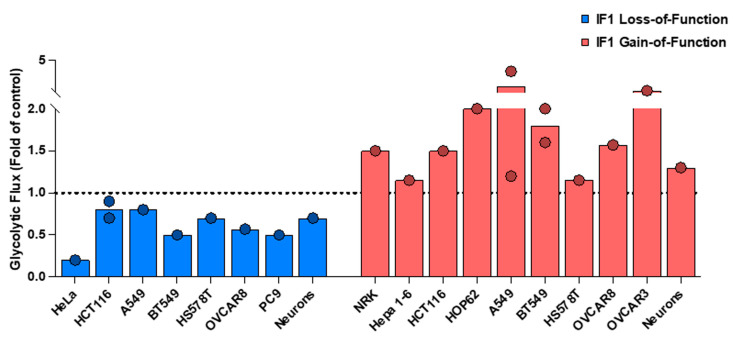
IF1 promotes metabolic reprogramming to an enhanced glycolytic flux. The histograms represent significant results (*p* < 0.05) of the glycolytic flux in different cancer and non-cancer (NRK, neurons) cells by silencing or knocking out IF1 and by its overexpression (transiently or stably transfected) when compared to parental cells. The dashed line set at a value of 1.0 fold denotes the lack of effect in the parameter as a result of IF1 loss- or gain-of-function. Data were derived from [2,3,4,5,12,23,24]. Primary neuronal cultures derived from the brains of transgenic mice overexpressing IF1 or with deleted IF1 are represented as compared to neurons of control mice [7]. All data were normalized to its respective control to make the different studies represented comparable.

## 3. The Impact of IF1 on ATP Synthesis: Uncovering Cellular Effects

It is well established that cellular ATP levels are maintained at a fixed value by variable contributions of glycolysis and OXPHOS, after silencing or overexpressing IF1 as documented (see Figure 1G in [2]). Sgarbi and colleagues used an end-point assay of cellular ATP content in IF1-silenced and non-silenced permeabilized cells and found no differences in ATP synthesis. Surprisingly, Figure 4 in [1] does not show the cellular ATP content assayed in cells in the presence of oligomycin or in the presence of 2-desoxiglucose, to respectively assess the relative contribution of OXPHOS and glycolysis to cellular ATP production. In other words, the lack of differences in ATP synthesis in their study most likely result because they determined cellular steady-state levels of ATP. It is likely that IF1-silenced cells produced more ATP by OXPHOS than by glycolysis to maintain constant ATP levels. Consistent with our previous findings [2], we have recently reported that knocking out IF1 in mouse colonocytes and CT26 and MC38 cancer cells triggers the upregulation of ATP turnover without affecting basal mitochondrial ATP levels [11].

To assess the effect of IF1 in the ATP synthetic activity of ATP synthase, the kinetic mode assay originally developed by Barrientos [25] and detailed as a Bio-Protocol by our group [26] would be more appropriate. The use of the kinetic mode assay to assess ATP synthase activity in the presence or absence of oligomycin provides clear evidence that IF1 is inhibiting ATP synthase under basal cellular conditions. This method was implemented using isolated mitochondria and permeabilized cancer and non-cancer cells in which the mitochondrial content of IF1 was manipulated by its silencing, by knocking it out, or by its overexpression, both in transiently and in stably transfected cells (Figure 2) [11,12,27,28,29]. More recent results, using permeabilized IF1-knockout HCT116 and Jurkat cells and isolated brain, kidney, heart, and colon mitochondria of genetically modified mouse models of loss-of-function of IF1 [7,11,12], confirmed that IF1 indeed inhibits a fraction of mitochondrial ATP synthase in vivo (Figure 2). Likewise, Figure 2 also summarizes the inhibitory effect of IF1 on ATP synthase activity in isolated mitochondria of mice overexpressing IF1 in neurons [7]. Similar inhibitory effects on ATP synthase are obtained when the more active IF1-H49K inhibitor is expressed in liver and skeletal muscle [9,10], which are tissues devoid of IF1 in mice [28], and thus naturally, lack a fraction of IF1-inhibited enzyme [12].

**Figure 2 cancers-16-01014-f002:**
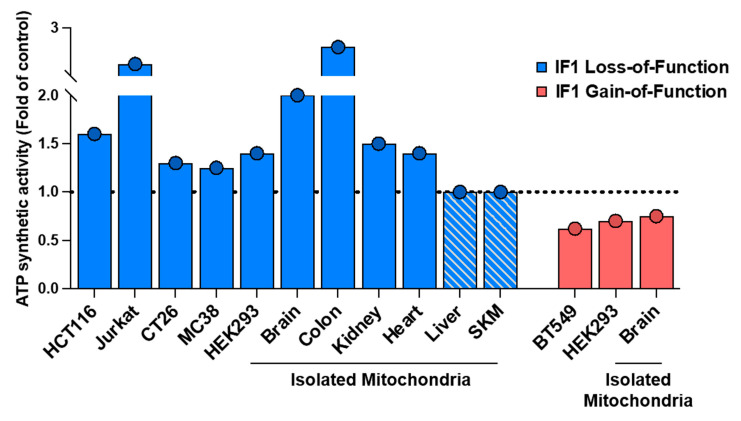
IF1 inhibits the ATP synthetic activity of ATP synthase. The histograms represent significant results (*p* < 0.05) of the ATP synthetic activity of mitochondrial ATP synthase in different cancer and non-cancer (HEK) cells by silencing or knocking out IF1 or by its overexpression (transiently or stably transfected) when compared to parental cells. The dashed line set at a value of 1.0 fold denotes the lack of effect in the parameter as a result of IF1 loss- or gain-of-function. Data were derived from [11,12,24]. ATP synthase activity determined in isolated mitochondria from IF1-ablated or IF1-overexpressing mouse tissues is also represented when compared to the activity determined in mitochondria of the corresponding control mice [7,11,12]. The hatched histograms in liver and skeletal muscle are used to emphasize that these tissues lack IF1 expression and hence are devoid of IF1-inhibited ATP synthase [12]. All data were normalized to its respective control to make the different studies represented comparable.

## 4. Loss- and Gain-of-Function of IF1 Impacts on Mitochondrial Respiration Rates

Traces of basal mitochondrial respiration assessed in a Clark-type electrode using permeabilized IF1-silenced and non-silenced cells were also used to support the lack of effect of IF1 in OXPHOS [1]. Surprisingly, the sequential traces of respiration in the presence of ADP, oligomycin, FCCP, and rotenone + antimycin were not shown, so that the reader cannot ascertain what is actually measured. Currently, there is some uncertainty surrounding the use of detached and permeabilized cells to assess a respiratory profile. Seahorse equipment with intact and attached cells would be more precise. The Seahorse technology solved, almost two decades ago, many of the problems that arise in experiments with detached (scraped/trypsinized) and permeabilized cells. Our cellular respiration studies, after the manipulation of IF1 content, were carried out in intact attached cells using the Seahorse technology [2,3,4,5,23,30], paying special attention to the effect on the oligomycin-sensitive respiration (OSR), which is a respiratory index of ATP synthase activity. Figure 3 summarizes the effect of IF1 dose on OSR in cancer and non-cancer cells when the mitochondrial content of IF1 is manipulated by its silencing, by knocking it out, or by its overexpression, both in transiently and stably transfected cells, as well as in primary neuronal cultures of genetically modified mouse models of loss- and gain-of-function of IF1 in the brain, clearly supporting a relevant inhibitory role for IF1 in ADP-stimulated mitochondrial respiration [2,3,5,7,9,11,12,23].

**Figure 3 cancers-16-01014-f003:**
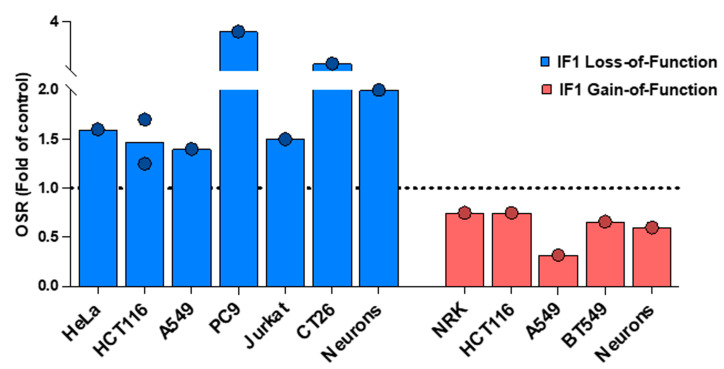
IF1 inhibits the oligomycin-sensitive respiration (OSR) of cancer and non-cancer cells. The histograms represent significant results (*p* < 0.05) of OSR in different cancer and non-cancer (neurons) cells by silencing or knocking out IF1 or by its overexpression (transiently or stably transfected) when compared to parental cells. The dashed line set at a value of 1.0 fold denotes the lack of effect in the parameter as a result of IF1 loss- or gain-of-function. Data were derived from [2,3,5,11,12,24]. Primary neuronal cultures derived from the brains of transgenic mice overexpressing IF1 or with deleted IF1 are represented when compared to neurons of control mice [7]. All data were normalized to its respective control to make the different studies represented comparable.

## 5. The Link between IF1 Dose and Mitochondrial Membrane Potential

Quantitative flow cytometry data of the mean fluorescence intensity of TMRM stained cells showed an increase (143B cells) or no differences (HCT116 and HeLa) in ΔΨm in IF1-silenced vs. parental control cells [1]. These results were used to question the implication of IF1 in ΔΨm [2,3,7,11,12,23]. However, the results do not show several controls that are required to assess the effect of IF1 in ΔΨm [1], such as the addition of an uncoupler (FCCP, DNP), to assess the specificity of mitochondrial TMRM staining, or of oligomycin, to assess the effect of the absence or presence of IF1 on ATP synthase in ΔΨm. Likewise, to rule out that TMRM signals after the manipulation of IF1 content are not affected by differences in the activity of multidrug resistance pumps, it is prudent to show the effect of cyclosporine H in ΔΨm. Figure 4 summarizes the effect of IF1 dose on ΔΨm in cancer and non-cancer cells when the mitochondrial content of IF1 is manipulated by its silencing, by knocking it out or by its overexpression, both in transient and in stable transfectants, as well as in primary neuronal cultures of genetically modified mouse models of loss- and gain-of-function of IF1 in forebrain neurons, supporting the participation of IF1 in the upregulation of ΔΨm.

Importantly, more recently, we demonstrated by STimulated Emission Depletion (STED) microscopy that an IF1-GFP construct that binds ATP synthase, determines cristae microdomains of high ΔΨm in live cells [12]. Remarkably, an IF1-S39-GFP mutant or GFP targeted to mitochondria, which are not able to bind ATP synthase, do not co-localize with cristae microdomains of high ΔΨm, further supporting a prominent role for IF1 in the regulation of ATP synthase in vivo and in the control of ΔΨm [12].

**Figure 4 cancers-16-01014-f004:**
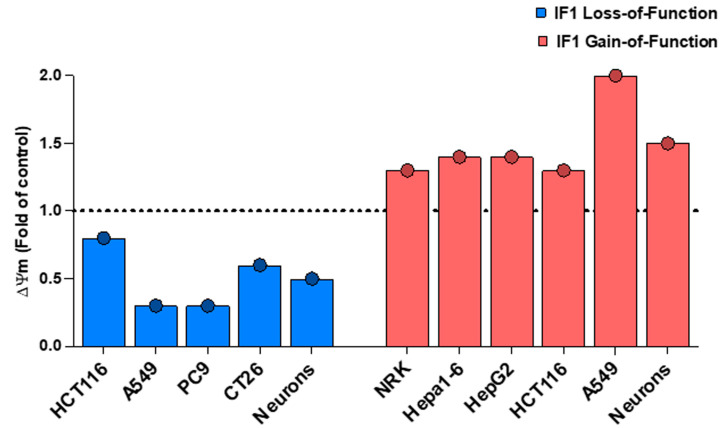
IF1 increases the mitochondrial membrane potential (ΔΨm) in different cancer and non-cancer cells. The histograms represent significant results (*p* < 0.05) of ΔΨm in different cancer and non-cancer (NRK, neurons) cells by silencing or knocking out IF1 or by its overexpression (transiently or stably transfected) when compared to parental cells. The dashed line set at a value of 1.0 fold denotes the lack of effect in the parameter as a result of IF1 loss- or gain-of-function. Data were derived from [3,5,11,12]. Primary neuronal cultures derived from the brains of transgenic mice overexpressing IF1 or with deleted IF1 are represented when compared to neurons of control mice [7]. All data were normalized to its respective control to make the different studies represented comparable.

Overall, to assess the effect of IF1 in metabolic reprogramming on the activity of ATP synthase, cellular respiration, ΔΨm, mtROS production, and so on, experiments need to show and incorporate the appropriate controls and the use of methodologies that prevent excessive non-required manipulation of cells. Moreover, performing experiments on silencing of IF1 in cells in culture is adequate, but it is a limited approach and the experiments should be accompanied by other loss- and gain-of-function experiments of IF1, either in transiently or in stably transfected cells (Figures 1–5). Moreover, the findings from cells in culture do not necessarily have to reproduce the results obtained in vitro with recombinant proteins [31,32,33]. The latter provide clues to basic mechanisms, but lack the much greater complexity of the mechanisms operating in a cell in vivo.

Importantly, the overexpression and silencing/deletion of genes in cells in culture usually provide helpful hints of the action of the gene. However, special care should be taken with the nuclear encoded *ATP5IF1* gene. The IF1 protein is differentially expressed in highly proliferative cancer and non-cancer cells [3,23,30], its phosphorylation state and activity as an inhibitor of ATP synthase depends on the cell cycle phase [27], it is re-expressed when differentiated cells are reprogrammed into iPSC [34], it has a very short half-life [23] when compared to other subunits of the enzyme [35], and it is tissue- and species-specifically expressed in differentiated cells of mammalian tissues [28], emphasizing the complexity of its regulation and activities. These results stress the need to develop genetically modified mice to confirm the cell type-specific actions of IF1 in mitochondrial physiology in vivo. These mice can be further exploited as preclinical models to investigate the relevance of IF1 in human pathophysiology and its potential as a target for therapeutic intervention. These are some of the reasons why we generated transgenic mice with regulated tissue-specific expression of IF1, or of its mutant IF1-H49K, in different mouse tissues [6,7,8,9,10]. Moreover, it explains why we have developed the floxed-IF1 mouse that has allowed knocking-out IF1 globally [12], in neurons [7], colonocytes [11], and in specific lymphoid cells (manuscript in preparation). Our experiments in genetically modified mouse models of loss- and gain-of-function of IF1 support its function as an inhibitor of a fraction of mitochondrial ATP synthase under basal in vivo cellular conditions affecting metabolic reprograming, the activity of ATP synthase, ΔΨm and mtROS production, among other activities, as we previously demonstrated in cancer and non-cancer cells [2,3,23,30] and which was recently confirmed by an independent laboratory in a mouse model in heart [17].

## 6. PKA-Dependent Phosphorylation of IF1 on ATP Synthase Activity

We have previously demonstrated that the phosphorylation of IF1 in S39 prevents its interaction with ATP synthase and hence, its inhibitory activity on ATP synthase [27]. Moreover, we documented that cancer cells and prevalent human carcinomas [27] and mouse tissues that express IF1 [27,28] contain a variable amount of phosphorylated and dephosphorylated IF1. The phosphorylation of IF1 is prevented in cells in culture with inhibitors of PKA and stimulated by db-cAMP in cell lines that contain dephosphorylated IF1 [27]. Moreover, we reported that β-adrenergic stimulation promotes the phosphorylation and inactivation of IF1 to increase ATP synthase in vivo in mouse heart [27]. These results led us to suggest that IF1 is phosphorylated by PKA or a mitochondrial PKA-like activity [27].

To assess whether PKA-dependent phosphorylation of IF1 affects its inhibitory action on ATP synthase, Sgarbi and colleagues [1] treated control and IF1-silenced cells with H89, a competitive inhibitor of PKA, or with db-cAMP, a membrane permeable PKA activator, and assessed the rates of ATP synthesis and oxygen consumption by the same methodology mentioned previously. They reported no differential effect of db-cAMP in ATP synthesis and OCR between control and IF1-silenced cells in any of the three cell lines studied (Figures 7 and 9 in [1]). The lack of activation of ATP synthesis and of OCR by treatment with db-cAMP in HeLa and HCT116 cells could be explained because these cell lines already have all their IF1 in a phosphorylated state (see Figure 1 in [27]), that is, unable to interact with ATP synthase [27]. Therefore, no differential effect is expected to occur by the lack of IF1 because the phosphorylated IF1 provides a respiratory phenotype similar to that of cells depleted of inhibitors. We do not know the phosphorylation state of IF1 in 143B cells, presumably phosphorylated as in HCT116 and HeLa cells, since there is no OXPHOS response.

In agreement with previous results [27], Sgarbi et al. report a significant inhibition of ATP synthesis and oxygen consumption rates in the three cancer cell lines studied when using the H89 inhibitor of PKA [1]. At variance with a clear positive effect of IF1 silencing on the OSR and concurrent activation of the rates of aerobic glycolysis that we reported in H89-treated HCT116 cells (Figure 5) [27], these authors report that the silencing of IF1 has no effect on ATP synthesis and oxygen consumption rates [1], perhaps because of the methodological limitations used to assess ATP synthesis and oxygen consumption rates as pointed out here in previous sections. Moreover, they report that the effect of H89 on mitochondrial ATP synthesis and respiration is not affected when the electrons feed the respiratory chain at the level of complex II, thus limiting the effect of cAMP-dependent protein phosphorylation to complex I proteins of OXPHOS (Figure 8 in [1]). These findings contrast with other reports that have shown cAMP-dependent phosphorylation of other proteins of OXPHOS [27,36,37], the participation of additional kinases [38,39], and mechanisms of regulation of IF1 in response to adrenergic signaling [29]. In this regard, it should be noted that the third generation β-blocker nebivolol, that prevents the phosphorylation of NDUFS7 and the activity of complex I [29], also inhibits ATP synthase by sharply increasing the mitochondrial content of IF1, not by affecting its phosphorylation state [29]. Overall, showing the phosphorylation state of IF1 in this set of experiments would have helped their argument, specifically because they invoke phosphorylation/dephosphorylation of IF1 and its interaction with the enzyme without showing any experimental approach to verify it, which are usually requested experiments to build a constructive argument in the field [27].

**Figure 5 cancers-16-01014-f005:**
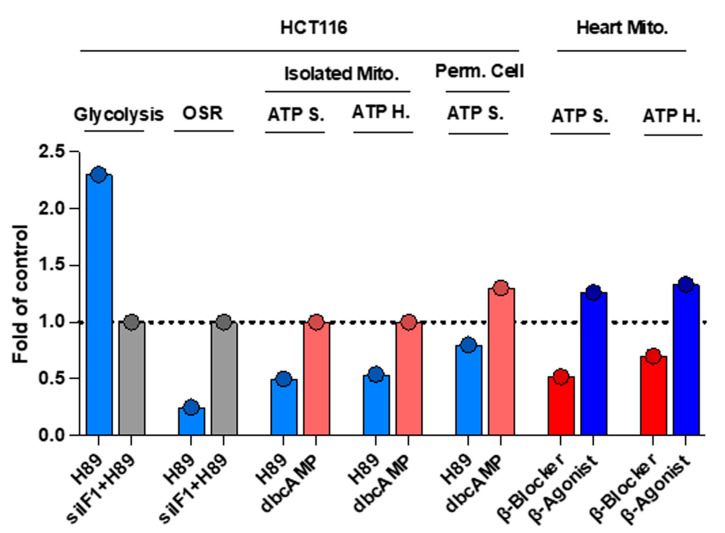
Inhibition of PKA by H89 inhibits ATP synthase and hydrolase activities of ATP synthase: In vivo effects by β-adrenergic signaling. The histograms represent significant results (*p* < 0.05) of the rates of aerobic glycolysis, OSR, ATP synthase (ATP S.) and ATP hydrolase (ATP H.) in isolated mitochondria of HCT116 cells and of ATP synthase (ATP S.) in permeabilized HCT116 cells after treatment with the PKA inhibitor H89 or the PKA activator db-cAMP. Moreover, the histograms also show the ATP synthase (ATP S.) and ATP hydrolase (ATP H.) activities of ATP synthase determined in isolated heart mitochondria after treatment of mice with the β-adrenergic blocker propranolol or the β-agonist clenbuterol when compared to data derived from non-treated mice (*p* < 0.05). The dashed line set at a value of 1.0 fold denotes the lack of effect in the parameter as a result of IF1 loss- or gain-of-function. Data were derived from [27]. All data were normalized to its respective control to make the different studies represented comparable.

## 7. PKA-Dependent Phosphorylation of IF1 on ATP Hydrolase Activity

An indirect experimental approach to assess the hydrolytic activity of ATP synthase in permeabilized cells was presented to argue against the role of PKA in controlling the hydrolytic activity of ATP synthase (Figure 10 in [1]). The method determines the steady-state content of ATP in permeabilized cells after mitochondrial uncoupling with FCCP in the presence of different PKA effectors. It was argued that direct spectrophotometric evaluation of its activity by NADH decay could not be used due to interference from many dehydrogenases [1]. With this approach, they reported that cellular ATP levels were not significantly modified by H89 or db-cAMP in IF1-silenced cells when compared to the parental ones. As previously commented, silencing of IF1 does not modify cellular ATP levels [2,11], but significantly increases the rates of oligomycin-sensitive respiration (Figure 3), ATP synthetic (Figure 2) and hydrolytic (Figure 6) activities of ATP synthase, mitochondrial ATP turnover [11], and concurrently decreases the rates of aerobic glycolysis (Figure 1), strongly questioning the methodology used in [1], because they have not assessed the contribution of glycolysis and of OXPHOS to cellular levels of ATP in the presence of both regulators of PKA. Moreover, their own results (Figure 10 in [1]) indicated significant differences of the effect of H89 in ATP content between the H89-treated IF1-silenced cells.

It is not clear to us why they decided to use the steady state levels of ATP in uncoupled permeabilized cells as an index of the ATP hydrolase activity of ATP synthase instead of directly determining the hydrolase activity of the enzyme in isolated mitochondria using the classic spectrophotometric method that follows NADH oxidation coupled to ATP hydrolysis [25,26]. With isolated mitochondria, the marginal NADH oxidation exerted by dehydrogenases can be monitored (see panel d in Figure 1 in [12]) and the specificity of H89 on ATP hydrolysis by mitochondrial ATP synthase is neatly documented by the partial arrest of NADH oxidation when compared to non-treated cells (see panel D in Figure 2 in [27]). Moreover, the addition of oligomycin before exhaustion of NADH in the reaction, clearly shows the arrest of the activity of ATP hydrolase (see panel d in Figure 1 in [12]).

**Figure 6 cancers-16-01014-f006:**
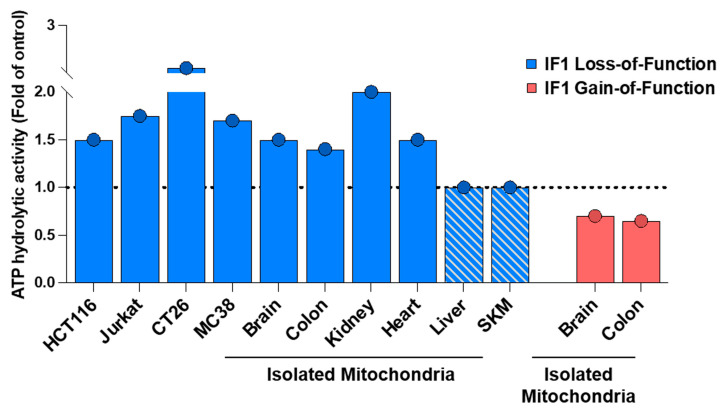
IF1 inhibits the ATP hydrolytic activity of ATP synthase. The histograms represent significant results (*p* < 0.05) of the ATP hydrolytic activity of ATP synthase in isolated mitochondrial of different cancer cells by knocking out IF1 when compared to parental cells. The dashed line set at a value of 1.0 fold denotes the lack of effect in the parameter as a result of IF1 loss- or gain-of-function. Data were derived from [11,12]. Likewise, the ATP hydrolytic activity of the enzyme determined in isolated mitochondria from the indicated mouse tissues derived from IF1-ablated or overexpressing IF1 is also represented when compared to the activity determined in mitochondria of the corresponding control mice [7,11,12]. The hatched histograms in liver and skeletal muscle are used to emphasize that these tissues lacking IF1 expression are devoid of IF1-inhibited ATP synthase [12]. All data were normalized to their respective control to make the different studies represented comparable.

More significantly, there are concerns about the validity of the argument presented to challenge our findings regarding the impact of H89 on ATP synthase activity, as these “were mostly based upon indirect assays, evaluating the rate of oligomycin-sensitive respiration (OSR) under basal cell culture conditions” [1]. This statement is incorrect because in addition to OSR, and in the same publication [27], we used the classic spectrophotometric method for the determination of ATP hydrolase activity in isolated mitochondria (see panel D in Figure 2 in [27]), as well as the ATP synthase activity in isolated mitochondria and in permeabilized cells supplemented with succinate as respiratory substrate (see panels E and F in Figure 2 in [27]). Moreover, we illustrated the co-immunoprecipitation of IF1 with the β-subunit of ATP synthase using two different inhibitors of the activity of PKA (H89, PKI) and two different anti-ATP synthase antibodies. Furthermore, treatment of the cells with db-cAMP prevented the co-immunoprecipitation of IF1 and ATP synthase (see panels A and B in Figure 2 in [27]). In addition, in the same figure we showed that H89 increased the content of IF1 co-fractionating with ATP synthase in Blue-native-gels (see panel C in Figure 2 in [27]). Finally, in the same study, we showed the sharp reduction of aerobic glycolysis in H89-treated cells upon the silencing of IF1 [27] (Figure 5). Overall, these data strongly supported the implication that PKA controls the activity of ATP synthase mediated by the phosphorylation of IF1 [27].

Moreover, they omitted quoting the results obtained in ATP synthetic (Figure 2) and hydrolytic activities of ATP synthase (Figure 6) reported in isolated mitochondria of IF1 mouse models of loss- and gain-of-function of IF1 that clearly demonstrated that IF1 is an inhibitor of both activities of the enzyme operating under basal physiological conditions [7,11,12]. Interestingly, that is not the case in liver and skeletal muscle mitochondria (Figure 6) [12], because these mouse tissues are devoid of IF1 expression [28]. Similarly, as recently documented [11,12], knocking out IF1 in cancer cells also triggers a sharp increase in the hydrolytic activity of ATP synthase in full agreement with our overall postulates regarding the activity of IF1 as a regulator of ATP synthase activity under non-stressed physiological in vivo conditions (Figure 6).

Regarding the effect of PKA they concluded that (i) activation or inhibition of PKA cannot affect the action of IF1 and its modulation of ATP synthase activities and (ii) IF1 does not inhibit the ATP synthesis rate via OXPHOS in cancer cells. However, we have documented the differential phosphorylation of IF1 in serine residues in cancer cells and in prevalent human carcinomas [27], and in different mouse tissues [28], and provided clear evidence that cAMP-dependent phosphorylation of S39 in IF1 by a PKA-like activity prevents the interaction of the inhibitory protein with the enzyme, whereas the dephosphorylated IF1 interacts and inhibits both ATP synthesis and hydrolysis by ATP synthase [27]. We have used different experimental approaches such as respiratory profiles by Seahorse technology in attached cells, co-immunoprecipitations, Blue-native gels, documented the effect of oligomycin in the determination of ATP synthase and hydrolase activities of the enzyme, and use of the corresponding phosphomimetic and phosphodeficient serine mutants of IF1 to support our findings [27]. Moreover, we have illustrated the differential phosphorylation of IF1 in: (i) the regulation of the metabolic flux of cancer cells that are forced into glycolysis or into OXPHOS, (ii) the transition into the OXPHOS-dependent and glycolytic-dependent phases of cells during their progression through the cell cycle, (iii) in the adaptation of the cells to hypoxia, and (iv) that the majority of IF1 is dephosphorylated, and hence active as an inhibitor of ATP synthase, in human breast, colon, and lung carcinomas [27]. More importantly, we have shown that clenbuterol, an activator of β-adrenergic signaling, when administered in vivo to mice, promotes cAMP accumulation in heart mitochondria and the phosphorylation of IF1 with the concurrent activation of both the synthetic and hydrolytic activities of ATP synthase in isolated heart mitochondria [27]. Contrariwise, the administration of the β-blocker propranolol had no relevant effect in cAMP levels but promoted the dephosphorylation of IF1 and the concurrent inhibition of both activities of ATP synthase stressing, by a pharmacologic approach, the existence of a fraction of IF1-inhibited ATP synthase in the in vivo heart under physiological conditions [27]. Finally, since the existence of mitochondrial PKA is still a subject of debate, we supported the action of a PKA-like activity, without ruling out the participation of other S/T kinases and phosphatases [38,39], to act as regulators that could promote the phosphorylation of IF1 in S39 to render the inactive inhibitor to allow the adaptation of cellular metabolism to changing cues.

In agreement with our results, the implication of PKA has also been claimed in the regulation of glucose-stimulated insulin secretion by IF1 [13,14] and in the IF1-promoted metabolic reprogramming experienced by epithelial cells upon exposure to carcinogens [15]. Moreover, blockade of the phosphorylation of IF1 after kynurenic acid signaling through GPR35 orphan receptors, has been recently reported to play a most prominent role in the oligomerization and inactivation of ATP synthase, to provide an anti-ischemic mechanism for ATP conservation in human and mouse cardiomyocytes [39]. Likewise, the lack of phosphorylation of IF1 in S39 also seems to be of relevance in the metabolic reprogramming to a Warburg phenotype of a set of High Grade Gliomas with IDH mutations [40]. Overall, these findings stress the relevance that the phosphorylation of IF1 in S39 plays in mitochondrial physiology and pathophysiology and emphasize the need of well documented studies to advance the field.

## 8. Tissue-Specific Duality of IF1 as Tumor Promoter or Tumor Suppressor

Finally, we should stress that, although we pioneered the idea of IF1 as a pro-oncogenic mitochondrial protein [41] promoting metabolic reprogramming of carcinomas to an enhanced glycolytic phenotype by the inhibition of ATP synthase [2,3,23], this terminology cannot be generalized for IF1 and should be used with caution. Indeed, the IF1-mediated inhibition of ATP synthase in addition to the promotion of metabolic reprogramming, further generated a mtROS signal in the respiratory chain that activated transcription factors such as NFκB that induced cellular proliferation and survival [2]. Certainly, a study in a large cohort of human hepatocarcinomas confirmed that IF1 overexpression is a biomarker of poor patient prognosis and that IF1 signaling through NFκB supports its pro-oncogenic role in hepatocarcinomas [42]. These results were further reinforced by the enhanced hepatocarcinogenesis observed in transgenic mice overexpressing the active IF1-H49K mutant in the liver [10]. Similarly, the overexpression of IF1 in bladder carcinomas and gliomas also predict a worse prognosis for patients [43,44] and the silencing of IF1 diminished the proliferation and invasive capacity of the cells [42,43,44].

However, despite IF1 promotion of metabolic reprogramming of cancer and non-cancer cells to an enhanced glycolytic phenotype in all of the cellular types that we studied (Figure 1), including hMSC [30] and reprogrammed iPSC [34], the overexpression of IF1 in lung, breast, and colon carcinomas correlates with a good prognosis for patients [4,5,23,24], suggesting that in these specific tissues, IF1 might exert tumor suppressor functions. To investigate this point, we generated loss- and gain-of-function IF1 cells of breast, colon, and lung carcinomas [4,5,24]. Definitely, IF1, despite favoring glycolysis, triggered a more epithelial type transcriptome when compared to IF1-ablated or knockdown cells [4,5,24]. Functionally, breast, colon, and lung cancer cells expressing IF1 proliferate, migrate, and invade less than those expressing low levels or devoid of IF1 [4,5,24]. Moreover, cells with high IF1 levels are prone to anoikis, a form of cell death upon cellular detachment [4,5], and the tumor spheroids that they develop show an increased cell death by enhanced immune surveillance by Natural Killer cells [4]. Altogether, IF1-ablated colon and lung cancer cells are more tumorigenic and metastatic in vivo than IF1-expressing cells [4,5]. As we have recently suggested [45], the tumor suppressor role of IF1 in these types of carcinoma should stem from the differential nuclear reprogramming of kinases and transcription factors exerted by the mtROS generated in the respiratory chain as a result of the IF1-mediated inhibition of ATP synthase [2]. These are the reasons we currently oppose generalizing the idea of IF1 as a pro-oncogenic protein because its contribution to oncogenesis largely depends on the type of cell where the carcinoma develops. Hence, therapeutic approaches targeting IF1 should take into consideration its tissue-specific expression and behavior in oncogenesis.

## 9. Conclusions and Future Directions

Loss- and gain-of-function of IF1 studies in cancer, non-cancer cells, and in isolated mitochondria of different mouse tissues support that IF1 contributes to the Warburg effect by inhibiting a fraction of ATP synthase in the mitochondrion under aerobic conditions in vivo. Moreover, cAMP-dependent phosphorylation at S39 of IF1 by a mitochondrial PKA-like activity is required to render an inactive IF1 unable to bind the enzyme leading to the activation of ATP synthase. The phosphorylation of IF1 is of upmost importance to control the activity of OXPHOS and hence of metabolic reprogramming under physiological conditions, in adaptation to hypoxia and cancer. We support further studies in loss- and gain-of-function *Atp5if1* mouse models to advance the understanding of IF1 in human pathophysiology.

## Data Availability

The data can be shared up on request.

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
