# Peer review of "The ATPase Inhibitory Factor 1 (IF1) Contributes to the Warburg Effect and Is Regulated by Its Phosphorylation in S39 by a Protein Kinase A-like Activity"

_cancers, 2024, doi:10.3390/cancers16051014_

Round 1

Reviewer 1 Report

Comments and Suggestions for Authors

The manuscript by Cuezva and Domínguez-Zorita attempted to summarize their findings and those of others that favour the implication of IF1 in metabolic reprogramming which is mediated by its inhibitory activity on the ATP synthase. The authors also discuss and challenge the results that argue against the role of IF1 as in vivo inhibitor of mitochondrial ATP synthase.

The regulation of ATP synthase by IF1 is an essential aspect of cellular energy homeostasis. This dynamic interaction ensures that ATP synthase operates efficiently and adapts to the changing energy needs of the cell. Under physiological conditions, IF1 inhibits of ATP synthase during ATP surplus by binding to ATP synthase and resume its ATP synthesis activity during energy demand when cellular ATP levels decrease, by releasing from ATP synthase.  In the context of cancer cells, IF1 as an inhibitor of mitochondrial ATP synthase functions in the context of altered cellular metabolism commonly observed in cancer.

·         When manipulating IF1 levels using down- or up-regulation techniques to assess the effect of IF1 content on the rate of ATP synthesis, it is important to compare the same experimental method to reproduce the results.

·         When IF1 is silenced in stably transfected cell lines, they have already lost the ability to modulate ATP content depending on cellular demand and likely have a steady-state rate of ATP synthesis as measured in the controversial paper.

·         In the case of transient IF1 expression/silencing, cells may receive varying amounts of plasmids and therefore have an effect in terms of variability in the number of IF1 modulators per cell, which may be reflected in the rate of ATP synthesis.

Indeed, ATF production rates upon acute cell activation (increased ATP demand) showed almost three-fold increase in the overall rate of ATP turnover with the balance between OXPHOS and glycolysis broadly unchanged upon activation. This kind of activation minimally affect snapshot-in-time intracellular ATP levels, as cells will readily adapt to a new steady-state of increased ATP turnover.

Recent studies reports on the functional heterogeneity of mitochondria. Imaging analysis has shown that mitochondrial cristae display different membrane potentials within the same mitochondrion, implying differences in OXPHOS electron transfer chain activity and oxidative phosphorylation [Wolf, Individual cristae within the same mitochondrion display different membrane potentials and are functionally independent. EMBO J. 2019, 38, e101056]

Regarding metabolic reprogramming

In non-cancerous cells you used, when mitochondrial IF1 content is increased due to its overexpression, what type of metabolic reprogramming occurs and do the “reprogrammed” states persist after IF1 overexpression ceases?

Do the Warburg effect is observed?

To elucidate the effect of down- or up-regulated IF1 on the rate of ATP generation by glycolysis and oxidative phosphorylation, an approach of simultaneous measurements of extracellular acidification and oxygen consumption can be applied. This methodology allows direct comparison of rates of glycolytic and oxidative ATP production, comparing their relative contributions and how each varies across conditions.

This approach could be also helpful to estimate the impact of phosphorylated state of IF1 on the rates of glycolytic and oxidative ATP production.

Regarding phosphorylation of S39 by protein kinase A-like activity.

Since in vitro PKA phosphorylation assays did not show IF1 phosphorylation and cyclic AMP (cAMP)-dependent protein kinase A is a predicted kinase, the specificity of PKA for IF1 phosphorylation should be studied further, keeping in mind that no other author publication exists.

Minor

Line 39-42

The first sentence is duplicated.

Lane 109 typed ‘stablished’

Comments on the Quality of English Language

English is good, there are some typos.

Author Response

The manuscript by Cuezva and Domínguez-Zorita attempted to summarize their findings and those of others that favour the implication of IF1 in metabolic reprogramming which is mediated by its inhibitory activity on the ATP synthase. The authors also discuss and challenge the results that argue against the role of IF1 as in vivo inhibitor of mitochondrial ATP synthase.

The regulation of ATP synthase by IF1 is an essential aspect of cellular energy homeostasis. This dynamic interaction ensures that ATP synthase operates efficiently and adapts to the changing energy needs of the cell. Under physiological conditions, IF1 inhibits of ATP synthase during ATP surplus by binding to ATP synthase and resume its ATP synthesis activity during energy demand when cellular ATP levels decrease, by releasing from ATP synthase.  In the context of cancer cells, IF1 as an inhibitor of mitochondrial ATP synthase functions in the context of altered cellular metabolism commonly observed in cancer.

Response: We thank the reviewer for the brief and clear summary of the actions of IF1 on ATP synthase activity in response to ATP surplus or ATP demand supporting our arguments.   

When manipulating IF1 levels using down- or up-regulation techniques to assess the effect of IF1 content on the rate of ATP synthesis, it is important to compare the same experimental method to reproduce the results.

When IF1 is silenced in stably transfected cell lines, they have already lost the ability to modulate ATP content depending on cellular demand and likely have a steady-state rate of ATP synthesis as measured in the controversial paper.

In the case of transient IF1 expression/silencing, cells may receive varying amounts of plasmids and therefore have an effect in terms of variability in the number of IF1 modulators per cell, which may be reflected in the rate of ATP synthesis.

Indeed, ATF production rates upon acute cell activation (increased ATP demand) showed almost three-fold increase in the overall rate of ATP turnover with the balance between OXPHOS and glycolysis broadly unchanged upon activation. This kind of activation minimally affect snapshot-in-time intracellular ATP levels, as cells will readily adapt to a new steady-state of increased ATP turnover.

Response: We agree with these comments of the reviewer. Of course, we have always used the same methodology to assess the relevance of IF1 loss or gain of function on the rate of ATP synthesis/hydrolysis when compared to the corresponding controls. Moreover, in agreement with the reviewer’s comment, we have recently reported that knocking out IF1 in vivo in mouse colonocytes or in CT26 and MC38 colon cancer cells triggers a sharp increase in ATP turnover without affecting mitochondrial ATP levels (Cell Death Dis 2023, 14, 413). In any case, as we have recommended in section 3 of the manuscript, the kinetic mode to assess ATP synthesis rates in permeabilized cells or in isolated mitochondria in the absence/presence of oligomycin is a robust and better assay than the determination of steady-state levels of ATP.

Recent studies reports on the functional heterogeneity of mitochondria. Imaging analysis has shown that mitochondrial cristae display different membrane potentials within the same mitochondrion, implying differences in OXPHOS electron transfer chain activity and oxidative phosphorylation [Wolf, Individual cristae within the same mitochondrion display different membrane potentials and are functionally independent. EMBO J. 2019, 38, e101056]

Response: We acknowledge this comment of the reviewer. In fact, in section 5 we have summarized that the interaction of IF1 with ATP synthase is a main player in determining the heterogeneity of ΔΨm within cristae of the same mitochondrion in vivo, as we demonstrated by STED microscopy studies (Commun Biol 2023, 6, 836). In that contribution, we quoted the papers of Wolf et al., EMBO J. 2019, 38, e101056 and Rieger et al., EMBO Rep. 2021, 22, e52727 as pioneering studies regarding the heterogeneity of ΔΨm in the mitochondrion.

 Regarding metabolic reprogramming

In non-cancerous cells you used, when mitochondrial IF1 content is increased due to its overexpression, what type of metabolic reprogramming occurs and do the “reprogrammed” states persist after IF1 overexpression ceases?

Response: We have studied metabolic reprograming in normal rat kidney cells (NRK) by transient overexpression of IF1 and showed the partial inhibition of oligomycin sensitive respiration (OSR), the reprogramming to an enhanced glycolytic flux and increased ΔΨm (Figs. 3,1, 4). We have never studied what happens when IF1 expression ceases in this situation. Instead, the approach we have followed to assess metabolic reprogramming when IF1 activity ceases in non-cancer cells has been the study of the metabolic reprogramming experienced in neurons and colonocytes derived from mice with tissue-specific deletion of the Atp5if1 gene in forebrain neurons or intestinal epithelium, respectively. In these cells we have reported that ablation of IF1 enhances the activity of ATP synthase and mitochondrial respiration and promotes a decrease in glycolysis that parallels the content of IF1. Likewise, we have also studied metabolic reprograming in neurons, colonocytes and hepatocytes derived from mice with tissue-specific overexpression of hIF1, or of its mutant IF1-H49K, in forebrain neurons, intestinal epithelium or liver. The results invariably show that an increase in IF1 dose inhibits ATP synthase and mitochondrial respiration and promotes an increase in the flux of glycolysis and/or in the expression of glycolytic markers in the corresponding tissue.

Do the Warburg effect is observed?

Response: Yes, the IF1-mediated Warburg effect is observed both in cancer and in non-cancer cells.

To elucidate the effect of down- or up-regulated IF1 on the rate of ATP generation by glycolysis and oxidative phosphorylation, an approach of simultaneous measurements of extracellular acidification and oxygen consumption can be applied. This methodology allows direct comparison of rates of glycolytic and oxidative ATP production, comparing their relative contributions and how each varies across conditions.

This approach could be also helpful to estimate the impact of phosphorylated state of IF1 on the rates of glycolytic and oxidative ATP production.

Response: We agree with these comments of the reviewer which are the experimental approaches that we have used in most of our studies. However, since in the past the determination of the extracellular acidification rate (ECAR) was questioned as a surrogate approach to provide a direct estimation of the rate of aerobic glycolysis, we used in all of our studies the initial rates of lactate production for the estimation of aerobic glycolysis, including the experiments dealing with the phosphorylation state of IF1 (Cell Rep 2015,12, 2143-2155). In any case, in a recent contribution (Fig. 2 in PLoS Biol. 2021, 19, e3001252) we illustrate that ECAR and lactate production rates show parallel changes when IF1 is overexpressed or silenced in neurons.

Regarding phosphorylation of S39 by protein kinase A-like activity.

Since in vitro PKA phosphorylation assays did not show IF1 phosphorylation and cyclic AMP (cAMP)-dependent protein kinase A is a predicted kinase, the specificity of PKA for IF1 phosphorylation should be studied further, keeping in mind that no other author publication exists.

Response: We agree with the comment of the reviewer. There is an urgent need to further explore the signaling pathways and kinases/phosphatases that control the phosphorylation status of S39 that render IF1 as an inactive inhibitor of mitochondrial ATP synthase. Other studies have provided evidence that IF1 is phosphorylated in S39 in human skeletal muscle (Zhao et al., 2011, Mol Cell Proteomics 10, M110.000299) and cancer cell lines (Sharma et al., 2014, Cell Rep 8, 1583-1594; Zhou et al., 2013, J Proteome Res 12, 260-271) and that blockade of the phosphorylation of IF1 through GPR35 orphan receptor plays a most prominent role in the oligomerization and inactivation of ATP synthase, to provide an anti-ischemic mechanism for ATP conservation (Wyant et al., Science 2022, 377, 621-629).

 Minor

 Line 39-42: The first sentence is duplicated. Response: The duplicated sentence has been eliminated.

Lane 109 typed ‘stablished’. Response: The spelling of “established” has been corrected, now is in line 103.

Reviewer 2 Report

Comments and Suggestions for Authors

The manuscript entitled "The ATPase Inhibitory Factor 1 (IF1) contributes to the Warburg effect and is regulated by its phosphorylation in S39 by a protein kinase A-like activity" was submitted as a review by Cuezva and Domínguez-Zorita.

The topic of the review is of general interest. Moreover, the manuscript contains relevant information and several interesting aspects and ideas.

However, the manuscript has been prepared more or less like an original article. Each of the six figures displays a detailed original data analysis. This is not the idea behind a review. A review should present a summary and a generalization of the central results concerning a certain topic. This should give the reader the opportunity to gain an overview over the current state of the art and emerging concepts in a field.

There is another problem with the figures in this manuscript. The figure legends contain the information "data redrawn from", citing up to eight publications. It is not valid to combine the results of different studies published in different papers in one single figure. Even if these are experiments dealing with the same topic, it is not valid to give the impression that the results are directly comparable.  

Comments on the Quality of English Language

Moderate editing of English language required

Author Response

The manuscript entitled "The ATPase Inhibitory Factor 1 (IF1) contributes to the Warburg effect and is regulated by its phosphorylation in S39 by a protein kinase A-like activity" was submitted as a review by Cuezva and Domínguez-Zorita.

The topic of the review is of general interest. Moreover, the manuscript contains relevant information and several interesting aspects and ideas.

RESPONSE: We thank the reviewer for the positive comments towards our contribution.

However, the manuscript has been prepared more or less like an original article. Each of the six figures displays a detailed original data analysis. This is not the idea behind a review. A review should present a summary and a generalization of the central results concerning a certain topic. This should give the reader the opportunity to gain an overview over the current state of the art and emerging concepts in a field.

There is another problem with the figures in this manuscript. The figure legends contain the information "data redrawn from", citing up to eight publications. It is not valid to combine the results of different studies published in different papers in one single figure. Even if these are experiments dealing with the same topic, it is not valid to give the impression that the results are directly comparable.  

RESPONSE: There is a clear misunderstanding here. The figures are new, meaning that as such they have never been published. However, the figures are not presenting detailed original data. Following the idea of what a review should present, the figures show the recalculated data for a particular parameter obtained in the different experiments that we have carried out represented as a fold change of its corresponding control in that experiment. Data has been normalized to their respective control to make comparable the different studies represented. The dashed line in the figures set at a value of 1.0 fold denotes the lack of effect in the parameter as a result of IF1 loss or gain of function. We apologize for the misunderstanding and not having made clear this point. Following the reviewer’s comments, in the revised version of the manuscript all figure legends explicitly state these points in two sentences.

Comments on the Quality of English Language. Moderate editing of English language required.

RESPONSE: We have edited English language as requested.

Reviewer 3 Report

Comments and Suggestions for Authors

This review focusses on the role of IF1 for Warburg effect. I have the following comments.

  1. The authors should present the review in a more balanced way and avoid direct polemic statements. For this they should remove the reference to the discrepancy of their view to the other other publication from the abstract and beginning of introduction. The reasons for discrepant views should be discussed in the light of experimental data.

  1. I would also strongly recommend to alter the headings of paragraphs to decouple the review a bit more from the obvious tight link to the paper in Ref. 1. For a reader not familiar with Ref.1 this review is otherwise very difficult to read. Simply spell out your arguments or use the italic statements as potential headings with question mark. 
Comments on the Quality of English Language

None.

Author Response

Reviewer 3

This review focusses on the role of IF1 for Warburg effect. I have the following comments.

1. The authors should present the review in a more balanced way and avoid direct polemic statements. For this they should remove the reference to the discrepancy of their view to the other other publication from the abstract and beginning of introduction. The reasons for discrepant views should be discussed in the light of experimental data.

RESPONSE: Following the reviewer’s request, we have ameliorated our statements to present our review in a more balanced way. Moreover, we have eliminated the alluded reference from the abstract and the beginning of the introduction.

2. I would also strongly recommend to alter the headings of paragraphs to decouple the review a bit more from the obvious tight link to the paper in Ref. 1. For a reader not familiar with Ref.1 this review is otherwise very difficult to read. Simply spell out your arguments or use the italic statements as potential headings with question mark. 

RESPONSE: We thank the reviewer for his/her valuable suggestion. Following the reviewer’s comment, we have changed the headings contributing to a clearer and more accessible structure for the review.

Comments on the Quality of English Language                        None.

Reviewer 4 Report

Comments and Suggestions for Authors

The ATPase Inhibitory Factor 1 (IF1) plays a prominent role as an inhibitor of mitochondrial ATP synthase. A recent publication claims that IF1 does not contribute to the Warburg effect and is not regulated by PKA in cancer cells. Contrary to recent claims, the authors reviewed the role of IF1 in promoting the Warburg effect and the relevance of its phosphorylation by PKA to block its activity as inhibitor of ATP synthase. They emphasize the phosphorylation of IF1 at S39 by a mitochondrial c-AMP-dependent PKA activity, that prevents its interaction with ATP synthase. Moreover, therapeutic approaches targeting IF1 should take into consideration its tissue-specific expression and behavior in oncogenesis. Overall, it is a well written article. The arguments from previous studies are logically discussed. The manuscript may be accepted for publication.

Author Response

The ATPase Inhibitory Factor 1 (IF1) plays a prominent role as an inhibitor of mitochondrial ATP synthase. A recent publication claims that IF1 does not contribute to the Warburg effect and is not regulated by PKA in cancer cells. Contrary to recent claims, the authors reviewed the role of IF1 in promoting the Warburg effect and the relevance of its phosphorylation by PKA to block its activity as inhibitor of ATP synthase. They emphasize the phosphorylation of IF1 at S39 by a mitochondrial c-AMP-dependent PKA activity, that prevents its interaction with ATP synthase. Moreover, therapeutic approaches targeting IF1 should take into consideration its tissue-specific expression and behavior in oncogenesis. Overall, it is a well written article. The arguments from previous studies are logically discussed. The manuscript may be accepted for publication.

RESPONSE: We thank the reviewer for the positive comments towards our review.

Round 2

Reviewer 3 Report

Comments and Suggestions for Authors

The authors have addressed all of my comments accordingly.